# Atypical Teratoid/Rhabdoid Tumor in Taiwan: A Nationwide, Population-Based Study

**DOI:** 10.3390/cancers14030668

**Published:** 2022-01-28

**Authors:** Yen-Lin Liu, Min-Lan Tsai, Chang-I Chen, Noi Yar, Ching-Wen Tsai, Hsin-Lun Lee, Chia-Chun Kuo, Wan-Ling Ho, Kevin Li-Chun Hsieh, Sung-Hui Tseng, James S. Miser, Chia-Yau Chang, Hsi Chang, Wen-Chang Huang, Tai-Tong Wong, Alexander T. H. Wu, Yu-Chun Yen

**Affiliations:** 1Department of Pediatrics, School of Medicine, College of Medicine, Taipei Medical University, Taipei 110, Taiwan; yll.always@gmail.com (Y.-L.L.); minlan456@hotmail.com (M.-L.T.); santeho@gmail.com (W.-L.H.); changtiang@yahoo.com.tw (C.-Y.C.); jamesc@tmu.edu.tw (H.C.); 2Department of Pediatrics, Taipei Medical University Hospital, Taipei 110, Taiwan; 3Pediatric Brain Tumor Program, Taipei Cancer Center, Taipei Neurological Institute, Taipei Medical University, Taipei 110, Taiwan; b001089024@tmu.edu.tw (H.-L.L.); kevinh9396@gmail.com (K.L.-C.H.); jamesmiser@gmail.com (J.S.M.); ttwong99@gmail.com (T.-T.W.); 4TMU Research Center of Cancer Translational Medicine, Taipei Medical University, Taipei 110, Taiwan; 5Department of Health Care Administration, College of Management, Taipei Medical University, Taipei 110, Taiwan; dcchen@tmu.edu.tw (C.-I.C.); m911108018@tmu.edu.tw (N.Y.); b8601093@tmu.edu.tw (C.-C.K.); 6Health Data Analytics and Statistics Center, Office of Data Science, Taipei Medical University, Taipei 110, Taiwan; s87120570@gmail.com; 7Department of Radiation Oncology, Taipei Medical University Hospital, Taipei 110, Taiwan; 8Department of Radiology, School of Medicine, College of Medicine, Taipei Medical University, Taipei 110, Taiwan; 9Ph.D. Program for Cancer Molecular Biology and Drug Discovery, College of Medical Science and Technology, Taipei Medical University and Academia Sinica, Taipei 110, Taiwan; 10Department of Radiation Oncology, Wan Fang Hospital, Taipei Medical University, Taipei 110, Taiwan; 11School of Medicine, College of Medicine, Fu Jen Catholic University, New Taipei City 242, Taiwan; 12Department of Pediatrics, Shin Kong Wu Ho-Su Memorial Hospital, Taipei 111, Taiwan; 13Department of Medical Imaging, Taipei Medical University Hospital, Taipei 110, Taiwan; 14Department of Physical Medicine and Rehabilitation, Taipei Medical University Hospital, Taipei 110, Taiwan; m003089010@tmu.edu.tw; 15Department of Physical Medicine and Rehabilitation, School of Medicine, College of Medicine, Taipei Medical University, Taipei 110, Taiwan; 16Department of Pediatrics, City of Hope Comprehensive Cancer Center, Duarte, CA 91010, USA; 17Cancer Center, Taipei Medical University Hospital, Taipei 110, Taiwan; 18Department of Pathology, Wan Fang Hospital, Taipei Medical University, Taipei 110, Taiwan; 99364@w.tmu.edu.tw; 19Graduate Institute of Clinical Medicine, College of Medicine, Taipei Medical University, Taipei 110, Taiwan; 20Neuroscience Research Center, Taipei Medical University Hospital, Taipei 110, Taiwan; 21Division of Pediatric Neurosurgery, Department of Neurosurgery, Taipei Medical University Hospital, Taipei Neuroscience Institute, Taipei Medical University, Taipei 110, Taiwan; 22The Ph.D. Program of Translational Medicine, College of Medical Science and Technology, Taipei Medical University, Taipei 110, Taiwan; 23Clinical Research Center, Taipei Medical University Hospital, Taipei Medical University, Taipei 110, Taiwan; 24Graduate Institute of Medical Sciences, National Defense Medical Center, Taipei 114, Taiwan

**Keywords:** atypical teratoid/rhabdoid tumor, CNS tumors, pediatric cancer, survival outcome

## Abstract

**Simple Summary:**

Atypical teratoid/rhabdoid tumor (AT/RT) is a rare, highly malignant CNS neoplasm with poor prognosis. A retrospective population-based analysis of patients with the diagnosis of AT/RT, registered between 1999 and 2014 in Taiwan, showed that: (1) AT/RT had a higher prevalence in males, in children < 36 months of age, and at infratentorial sites; (2) older age (≥12 months), presence of the tumor in the supratentorial region, use of radiotherapy, chemotherapy, or both were associated with better prognosis compared to surgery or no treatment. These data represent a historical experience with AT/RT in Taiwan and may inform risk stratification and clinical trial design.

**Abstract:**

Background: Atypical teratoid/rhabdoid tumor (AT/RT) is a rare, highly aggressive embryonal brain tumor most commonly presenting in young children. Methods: We performed a nationwide, population-based study of AT/RT (ICD-O-3 code: 9508/3) in Taiwan using the Taiwan Cancer Registry Database and the National Death Certificate Database. Results: A total of 47 cases (male/female = 29:18; median age at diagnosis, 23.3 months (IQR: 12.5–87.9)) were diagnosed with AT/RT between 1999 and 2014. AT/RT had higher prevalence in males (61.70%), in children < 36 months (55.32%), and at infratentorial or spinal locations (46.81%). Survival analyses demonstrated that patients ≥ 3 years of age (*n* = 21 (45%)) had a 5y-OS of 41% (*p* < 0.0001), treatment with radiotherapy only (*n* = 5 (11%)) led to a 5y-OS of 60%, treatment with chemotherapy with or without radiotherapy (*n* = 27 (62%)) was associated with a 5y-OS of 45% (*p* < 0.0001), and patients with a supratentorial tumor (*n* = 11 (23%)) had a 5y-OS of 51.95%. Predictors of better survival on univariate Cox proportional hazard modeling and confirmed with multivariate analysis included older age (≥1 year), supratentorial sites, and the administration of radiotherapy, chemotherapy, or both. Gender had no effect on survival. Conclusion: Older age, supratentorial site, and treatment with radiotherapy, chemotherapy, or both significantly improves the survival of patients with AT/RT.

## 1. Introduction

Atypical teratoid/rhabdoid tumor (AT/RT) is a rare and highly malignant cancer of the central nervous system (CNS). AT/RT represents 1 to 2% of all pediatric CNS tumors [1,2,3,4] and is the most common CNS malignant tumor in children under 3 years of age [1,5]. In children under the age of 1, AT/RT accounts for 40 to 50% of CNS malignancies [2]. It is more prevalent in males and in children of European descent [6,7]. AT/RT is characterized by loss-of-function alterations in the *SMARCB1* gene on chromosome 22q11.2 in more than 95% of patients, with the remainder having mutations in *SMARCA4*, located on chromosome 19p13.2 [2,8,9,10]. AT/RTs have been found throughout the CNS, most commonly in the infratentorial region; their location may vary with age [2,11].

Pathologically, AT/RTs are embryonal tumors that have a rhabdoid morphology, as well as areas with primitive neuroectodermal, mesenchymal, and epithelial features [5]. Radiographically, AT/RT typically presents as a large, heterogeneous mass with varying degrees of necrosis, hemorrhage, and peritumoral edema, mostly within the CNS but sometimes along the cranial nerves or at the skull base [2].

AT/RTs are highly malignant in nature and are classified as Grade IV CNS tumors according to the World Health Organization (WHO) classification [12]. Even with intensive multimodality therapies, the prognosis of AT/RT is poor, with a 15–53% of survival rate at three years and a median survival of approximately 1 year [3,8,13,14]. Due to the rare occurrence of AT/RT, the optimal treatment has yet to be determined, and therapeutic approaches vary from institution to institution [1,13]. AT/RTs are most commonly managed using a multimodality treatment that includes surgery followed by chemotherapy, radiotherapy, high-dose chemotherapy with stem cell therapy (SCT), and intrathecal or intraventricular (IT/IVent) chemotherapy [2]. Although the extent of surgical resection has been proven to be associated with better outcomes, there is no universally accepted chemotherapy or radiotherapy regimen for AT/RT. Previous studies suggest that patients may have a longer disease-free survival with SCT [5]. To reduce the risk of neurocognitive toxicity in younger patients, radiotherapy may be delayed; however, this may affect the overall survival [2,15].

The objectives of this study were to provide information on demographic characteristics and treatment approaches to inform risk stratification and clinical trial design for future studies of AT/RT.

## 2. Materials and Methods

### 2.1. Study Design and Participants

We conducted a nationwide, population-based, retrospective cohort study. We retrieved cases from the Taiwan Cancer Registry Database [16,17] and the National Death Certificate Database with pathological diagnosis of AT/RT with ICD-O-3 code: 9508/3, between 1 January 1999 and 31 December 2014. A total of 47 patients were identified to form the study cohort. Extracted demographic and clinical data included age, sex, resident location, tumor site, type of treatment received, and year of diagnosis.

### 2.2. Statistical Analysis

The mean ± standard deviations for continuous variables and proportions for categorical variables were used to present the study cohort’s demographic and clinical characteristics. Distributions of tumor site and treatment types were compared between age groups, and the association between treatment and location of the tumor was also analyzed. Categorical variables were compared using Chi-square test or Fisher’s exact test if the expected values were not large enough for the Chi-square test. Continuous variables were compared using Student’s *t*-test and analysis of variance (ANOVA). The survival rates were calculated using the Kaplan–Meier method and were used to check the proportional hazard assumption. The survival curves of different groups were compared using log-rank tests. Univariate Cox’s proportional hazard models were used to estimate the relative risk (crude hazard ratio (HR)) associated with age, gender, treatment type, tumor location, and diagnosis year, followed by multivariate Cox’s proportional hazard modeling. All statistical analyses were performed using SAS software (Version 9.4). Two-sided *p* values of <0.05 were considered statistically significant.

## 3. Results

### 3.1. Demographic and Clinical Characteristics

The demographic and clinical characteristics of the study population are shown in Table 1. Of 47 enrolled patients with AT/RT, 29 were male (61.70%). The mean age of the patients was 66.87 (±109.32) months; 11 patients were younger than 12 months of age (23.40%), 15 were 12 to 35 months old (31.91%), and 21 were 36 months of age or older (44.68%). Regarding the tumor site, 46.81% of the tumors were in the infratentorial region or in the spine (*n* = 22), 29.79% were at an unspecified site (*n* = 14), and 23.40% were in the supratentorial region (*n* = 11). In this group, 24 patients received combined radiotherapy and chemotherapy (51.06%), 12 patients received chemotherapy only (25.53%), 6 patients received surgery only or no treatment (12.77%), and 5 patients received radiotherapy only (10.64%). In addition, 24 patients were diagnosed between 1999 and 2007 (51.06%), and 23 patients were diagnosed between 2008 and 2014 (48.94%). Nearly half of the patients (*n* = 21, 44.68%) were from Northern Taiwan, followed by Central Taiwan (*n* = 15, 31.91%) and then Southern Taiwan (*n* = 11, 23.40%).

### 3.2. Distribution of Tumor Site and Treatment across Age Groups

In Table 2, which compares the tumor site and treatment across age groups, we noted a trend of a higher prevalence of infratentorial/spinal tumors in younger patients (*n* = 15) and of supratentorial tumors in older patients (*n* = 8) (*p* = 0.082). Children younger than 3 years of age more commonly had surgery only or no treatment (*n* = 6 and 0), and fewer received treatment with chemotherapy and/or radiotherapy than children 3 years or older (*n* = 20 and 21) (*p* = 0.026).

We found no significant relationship between tumor site and treatment (Table 3; *p* = 0.6588).

### 3.3. Prognostic Factors

Kaplan–Meier analysis (Figure 1) showed that the survival probabilities of the patients who were aged ≥36 months (Figure 1a), whose tumor was located at a supratentorial site (Figure 1d), and who received radiotherapy (Figure 1e), were significantly higher (all *p* < 0.05 by log-rank test) than those of the other patients. Gender, residence location in Taiwan, and diagnosis year had no significant influence on survival (Figure 1b,c,f; all *p* > 0.05 by log-rank test). When analyzing survival, we found that all infants with AT/RT diagnosed at age <12 months (*n* = 11 (23%)) died within 18 months from diagnosis, while cases diagnosed at ages 12–35 months (*n* = 15 (32%)) had a 5-year overall survival probability (5y-OS) of 28%, and those diagnosed at the age of ≥36 months (*n* = 21 (45%)) had a 5y-OS of 41% (*p* < 0.0001). All cases treated with surgery only (*n* = 6 (13%)) died within 6 months; all cases treated with chemotherapy without radiotherapy (*n* = 12 (25%)) died within 3 years; all cases treated with radiotherapy only (*n* = 5 (11%)) had a 5y-OS of 60%; the other cases treated with radiotherapy and chemotherapy (*n* = 24 (51%)) had a 5y-OS of 42.22% (*p* < 0.0001). Patients with a supratentorial tumor (*n* = 11 (23%)) had a 5y-OS of 51.95%, those with an unspecified nervous system tumor (*n* = 14 (30%)) had a 5y-OS of 21.43%, and those with an infratentorial or spine tumor (*n* = 22 (47%)) had a 5y-OS of 17.36% (*p <* 0.05).

Table 4 shows the results of univariate and multivariate analyses of factors that affect survival. Compared to the age group 0–11 months as a reference, the age groups 12–23 months (HR 0.113, 96% CI 0.039–0.330) and ≥36 months (HR 0.078, 95% CI 0.028–0.216) had a better prognosis on univariate analysis (both *p <* 0.001). On multivariate analysis, the 12–23 months group had a better prognosis (HR 0.130, 95% CI 0.036–0.468; *p* = 0.002). Compared to supratentorial tumors, tumors in the infratentorial or spine regions and tumors in an unspecified location of the nervous system or other location had a poorer prognosis in both univariate (HR 3.121, 95% CI 1.146–8.497; *p* = 0.026 and HR 3.261, 95% CI 1.139–9.337; *p* = 0.028 respectively) and multivariate analyses (HR 3.234, 95% CI 1.049–9.973; *p* = 0.041 and HR 3.505, 95% CI 1.121–10.955; *p* = 0.031). On univariate analysis, chemotherapy (including SCT) (HR 0.079, 95% CI 0.021–0.305; *p <* 0.001), radiotherapy (HR 0.011, 95% CI 0.002–0.063; *p <* 0.001), and combined radiotherapy and chemotherapy (HR 0.016, 95% CI 0.004–0.066; *p <* 0.001) were associated with better outcome. On multivariate analysis, chemotherapy (HR 0.013, 95% CI 0.002–0.097), radiotherapy (HR 0.002, 95% CI 0.000–0.025), and combined radiotherapy and chemotherapy (HR 0.003, 95% CI 0.000–0.031) remained significant protective prognostic factors (all *p <* 0.001). Gender and diagnosis year were not significant prognostic factors on either univariate or multivariate analysis.

## 4. Discussion

In this study, we identified 47 cases with a pathological diagnosis of AT/RT from 1999 to 2014. There was a higher proportion of males (61.70%), children younger than 36 months (55.32%), and patients with a tumor in an infratentorial or spinal location (46.81%). Older age at diagnosis, tumor in the supratentorial region, and treatment with radiotherapy with/without chemotherapy were found to be associated with better prognosis and 5-year overall survival. On the other hand, gender, residence location, and diagnosis year had no influence on prognosis.

Compared to other studies, we found a higher proportion of AT/RT in older patients (≥36 months) [7]. As a result, the median age at diagnosis (23.3 months) of our population appears to be higher than those reported in the Germany HIT database (14.4 months), by the Canadian consortium (16.7 months), and in the Austrian registry (17.28 months) [18,19,20], but similar to that reported in the United States’ registry [7] and clinical trials [21,22,23]. Improved overall survival with increasing age at diagnosis was reported by Katja et al. [23]; this is consistent with our results. In contrast, the Canadian Pediatric Brain Tumor Consortium found no survival advantage in children of an older age [18]; in their cohort, high-dose chemotherapy, a therapy associated with improved survival, was utilized in 42.9% of infants younger than 1 year of age compared to only one case in our study. Similar to our study, the SEER study and that by Julia et al. did not observe a differential risk in relation to gender; however, other studies reported either male or female predominance in the prevalence of AT/RT [6,24,25,26]. We also found that tumors in the supratentorial site have a better prognosis and 5-yr OS (51.95%) compared to tumors in other locations; this may be explained in part by the higher prevalence of supratentorial tumors in the ≥36 months age group. This finding is similar to those reported by the Japan Children Cancer Group [27], St. Jude Children’s Research Hospital [8] and Johann et al. [28] who defined and validated the molecular subgroups of AT/RT [28], in which the neurogenic (or MYC) subtype appeared to be more commonly associated with a supratentorial location and an older age at diagnosis.

Our study also confirmed the survival benefit of using radiotherapy for AT/RT treatment. In the U.S.A., evidence supporting patients’ long-term survival associated with radiotherapy has been reported in earlier studies [29,30,31,32], and the increased use of the radiotherapy in addition to surgical resection in recent years has been reported by Christine et al. [7]. Given the vulnerability of the developing brain to the adverse neurocognitive effects of radiation, radiotherapy has not been a standard treatment option in the younger patients. Severe long-term neurological, cognitive, and developmental effects, including a decline in intelligence quotient, were reported in patients who received radiation at a very young age [7,32,33]. Despite these risks, however, the results of our study demonstrate that radiotherapy is associated with the highest rate of survival (5y-OS of 60%) and should thus remain in the optimal treatment plan, whenever possible. Conformal focal radiation techniques including proton therapy, which allow for minimizing radiation exposure to normal brain tissue, have been successfully studied for other forms of CNS embryonal tumors [32,34,35] and have recently been utilized in AT/RT patients with reported favorable outcomes [29,36].

The optimal dose and volume of radiation therapy and the time of initial radiation therapy remain unclear [29]. A recent retrospective review from the Taipei Veterans General Hospital reported a significant association of high radiation dosage with better OS and progression survival (PFS), and delayed radiotherapy was associated with worse OS and PFS [30]. Early radiotherapy, within 2 months of diagnosis, was also significantly associated with better progression-free survival and reduction in overall mortality by 50% in children ≥3 years [4]. Our study did not include the details of radiotherapy such as dose, volume, and time of initiation in the analysis and was thus not able to provide their association with survival.

Innovations in chemotherapy intensity and regimens may improve the treatment outcomes. Several studies have also been conducted with the intention of studying whether SCT can delay irradiation and simultaneously preserve neurocognitive functions. A Vienna-based study used the strategy of delayed local irradiation after completion of chemotherapy with intensive intrathecal therapy for patients with localized disease and achieved excellent outcomes, reporting 100% OS, with only two patients relapsing among nine patients receiving intensive therapy [37]. The Children’s Oncology Group ACNS0333 study also reported improved survival with intensive postoperative chemotherapy and focal radiation therapy, with 43% OS and 37% event-free survival, respectively [15].

The role of chemotherapy has been extensively researched for the management of AT/RT. Given the severe neurocognitive side effects of radiotherapy, many patients receive chemotherapy in addition to surgical resection, with the aim of postponing or avoiding radiation therapy, especially in children under the age of 3 [7]. Though chemosensitive, AT/RT typically recurs within 6 months and progresses relentlessly [38]. An early study by Burger et al. reported that patients treated with chemotherapy only had a very poor prognosis, with most dying within 12 months. Favorable outcomes in AT/RT patients treated with radical surgery and aggressive chemotherapy have been reported [39]. Kai et al. studied high-dose chemotherapy with SCT and found that it may contribute to a better outcome [27]. Proteasome inhibitors (Marizomib, carfilzomib, and bortezomib) were recently studied as potential targeted therapy for patients with AT/RT, and tumor models have shown promising results [40,41].

There are some limitations in this study. Firstly, the diagnosis of AT/RT mainly relied on morphological diagnosis and immunohistochemistry with internal review in most pathology departments; central pathological reviews and molecular confirmation of *SMARCB1/SMARCA4* mutation/deletion status are not mandatory in Taiwan. It would be preferable for the histopathology to be subject to a confirmatory central review and to evidence of molecular analysis that supports the histopathological diagnosis; this is currently being established in Taiwan through collaborative efforts. The recent discovery of histopathological characteristics of AT/RT correlating with molecular subgroups determined by DNA methylation could further facilitate risk stratification and treatment planning in the future [42]. Secondly, details of the treatment, such as the extent of surgical resection, chemotherapy regimen, and radiotherapy dose, fields, and volume were not recorded in this dataset, and thus, we were not able to evaluate their impact on prognosis and survival. Although there have been more patients receiving high-dose chemotherapy in recent years, intrathecal chemotherapy and high-dose chemotherapy were not prevalent in Taiwan at the time of diagnosis and treatment of this population. Among the patients who received chemotherapy, only one patient in this database was recorded as having received a stem cell transplant. Consequently, we could not evaluate the effects of intrathecal chemotherapy or high-dose chemotherapy on prognosis and survival outcomes. Although cisplatin- or carboplatin-based chemotherapy regimens have been used in many centers [4], we did not yet have a standardized chemotherapy protocol for AT/RT in Taiwan during the study period. The recent discovery of the preclinical activity of proteasome inhibitors in AT/RT [40,41], however, has encouraged us to initiate a multi-center phase II trial using a proteasome inhibitor as an add-on therapy to standard chemotherapy for newly diagnosed AT/RT. Thirdly, since AT/RT is a rare cancer and the number of cases is limited, the analysis of the relationship between tumor site, treatment type, and age group required that small groups (of less than five patients) be combined as a single unit. Therefore, it was not possible to identify the association between each treatment type and the age groups. This was done in compliance with the rules of the database, to maintain patient confidentiality.

## 5. Conclusions

Since AT/RT is a rare disease, it is not easy for a single center to follow many patients longitudinally. The Taiwan Cancer Registry, a nationwide, population-based database, is therefore a useful resource for monitoring and analyzing the clinical characteristics and the treatment outcomes of AT/RT. We found that patients at an older age at diagnosis and those with supratentorial tumors had a better prognosis. Our data also support the effectiveness of radiotherapy, chemotherapy, or combined radiotherapy and chemotherapy. These data can inform future radiotherapy and chemotherapy regimens, clinical trial design, and risk stratification for AT/RT.

## Figures and Tables

**Figure 1 cancers-14-00668-f001:**
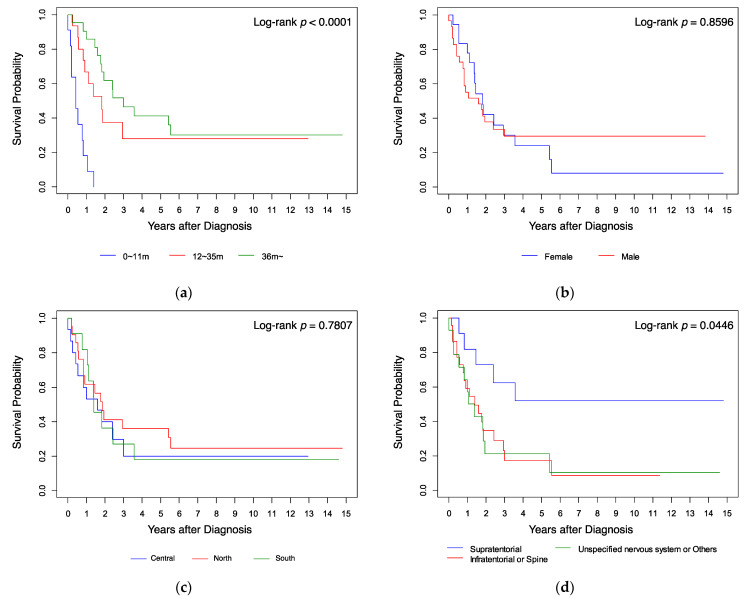
Kaplan–Meier analysis of survival curves compared by (**a**) age, (**b**) gender, (**c**) resident location, (**d**) tumor site, (**e**) treatment, and (**f**) diagnosis year.

**Table 1 cancers-14-00668-t001:** Demographic and clinical characteristics of the patients.

Variables	Total (*N* = 47) *
Mean age at diagnosis (months) * Median age at diagnosis (months)	66.87 (±109.32) 23.3 (12.5–87.9)
**Age group (months)**	
0–11 months	11 (23.40%)
12–35 months	15 (31.91%)
≥36 months	21 (44.68%)
**Gender, Male**	29 (61.70%)
**Residence Location**	
Northern Taiwan	21 (44.68%)
Central Taiwan	15 (31.91%)
Southern Taiwan	11 (23.40%)
**Tumor site**	
Supratentorial	11 (23.40%)
Infratentorial or Spine	22 (46.81%)
Unspecified nervous system or Others	14 (29.79%)
**Treatment**	
Surgery or no treatment	6 (12.77%)
Chemotherapy (including SCT)	12 (25.53%)
RT	5 (10.64%)
RT + CT	24 (51.06%)
**Diagnosis year**	
2008–2014	23 (48.94%)
1999–2007	24 (51.06%)

* Variables are expressed as mean ± standard deviation (SD) or median (interquartile range (IQR)) for continuous data, and *n* (%) for categorical data.

**Table 2 cancers-14-00668-t002:** Comparison of tumor site and treatment types for different age groups.

Tumor Site	Age < 3 Years (*N* = 26)	Age ≥ 3 Years (*N* = 21)	Total (*N* = 47)	*p* Value
Supratentorial	3 (27%)	8 (73%)	11	0.082 ^a^
Infratentorial or Spine	15 (68%)	7 (32)	22	
Unspecified nervous system or Others	8 (57%)	6 (43%)	14	
**Treatment**	**(*N* = 26)**	**(*N* = 21)**	**(*N* = 47)**	
Chemotherapy and/or radiotherapy	20 (49%)	21 (51%)	41	0.026 ^b^
Surgery or No treatment	6 (100%)	0 (0%)	6	

^a^ Chi-square test of independence; ^b^ Fisher’s exact test.

**Table 3 cancers-14-00668-t003:** Comparison of tumor site for different treatment types.

Tumor Site		Chemotherapy Only (*N* = 12)	Radiotherapy with/ without Chemotherapy (*N* = 29)	Surgery or No Treatment (*N* = 6)	Total (*N* = 47)	*p* Value *
Infratentorial or Spine		7 (32%)	12 (54%)	3 (14%)	22	0.698
Supratentorial, Unspecified or Others		5 (20%)	17 (68%)	3 (12%)	25	

* Fisher’s exact test.

**Table 4 cancers-14-00668-t004:** Univariate and multivariate analyses of factors affecting outcome.

Prognostic Factor	Univariate				Multivariate			
	HR	95% CI		*p* Value *	HR	95% CI		*p* Value *
**Age (months)**								
0–11 m		Reference				Reference		
12–23 m	0.113	(0.039–0.330)		**<0.001**	0.130	(0.036–0.468)		**0.002**
24–35 m	0.379	(0.082–1.754)		0.214	1.769	(0.175–17.852)		0.629
≥36 m	0.078	(0.028–0.216)		**<0.001**	0.356	(0.066–1.929)		0.231
**Gender**								
Female		Reference				Reference		
Male	0.942	(0.481–1.844)		0.862	1.012	(0.430–2.379)		0.979
**Tumor site**								
Supratentorial		Reference				Reference		
Infratentorial or Spine	3.121	(1.146–8.497)		**0.026**	3.234	(1.049–9.973)		**0.041**
Unspecified nervous system or Others	3.261	(1.139–9.337)		**0.028**	3.505	(1.121–10.955)		**0.031**
**Treatment**								
Surgery or No treatment		Reference				Reference		
Chemotherapy (including SCT)	0.079	(0.021–0.305)		**<0.001**	0.013	(0.002–0.097)		**<0.001**
Radiotherapy	0.011	(0.002–0.063)		**<0.001**	0.002	(0.000–0.025)		**<0.001**
Radiotherapy + Chemotherapy	0.016	(0.004–0.066)		**<0.001**	0.003	(0.000–0.031)		**<0.001**
**Diagnosis year**								
1999–2007		Reference				Reference		
2008–2014	1.439	(0.725–2.856)		0.298	1.316	(0.477–3.632)		0.596

* *p* values < 0.05 are presented in bold.

## Data Availability

The data presented in this study are available in the Taiwan Cancer Registry Database and in the National Death Certificate Database for researchers in Taiwan. The data are available at (https://dep.mohw.gov.tw/dos/cp-5119-59201-113.html; accessed on 26 December 2021) with the permission of the Ministry of Health and Welfare, Taiwan.

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
