# Peer review of "Atypical Teratoid/Rhabdoid Tumor in Taiwan: A Nationwide, Population-Based Study"

_cancers, 2022, doi:10.3390/cancers14030668_

Round 1

Reviewer 1 Report

It would be good to know the extent of surgical resection for the patient cohort.  In addition, the volume and extent of radiation would be helpful.  Both data sets would strengthen the conclusion of the paper.  I undertsand that this was not reportable (based on the comments in the discussion line 263-4).  Some details of chemotherapy regimens would also strengthen the paper.  For example was the main backbone of chemotherapy alkylator based.  The patient group diagnosed with AT/RT was older than seen in other reports, and had a sex preponderance (male/female).  Diagnostic details need to be included; was it solely on histopathology, were the histopathology slides reviewed and was there histopathological agreement.  Was the histology also paired with molecular analysis (SMARCB1 and SMARCA4 mutations) to be certain that the diagnosis was AT/RT.

The authors correctly identify the diagnosis of AT/RT is based on histopathology and moelcualr anayiss of SMARCB1 and SMARCA4 mutations.  It would be preferable for the histopathology to be subject to confirmatory central review, and for evidence of molecular analsysis that supports the histopathological diagnosis.  There are 3 molecular subgroups of AT/RT and a recent paper has described histopathological characteristics correlating with these groups {Zin F, Cotter J A et al Brain Pathol 2021 Sep; 31 (5)}.  Molecular subgrouping may also reflect prognosis and this needs to be confirmed by DNA methylation.  

Reviewer 2 Report

The authors present a retrospective population-based analysis of patients diagnosed with atypical teratoid/rhabdoid tumor (AT/RT), a rare, highly malignant CNS tumor. The paper is well written, presents interesting results about a very uncommon disease, and limits of the study design are clearly stated.

The authors found that overall survival improves with increasing age at diagnosis. However, as they say, the Canadian Pediatric Brain Tumor Consortium found no survival advantage in older children. Can the authors give some possible explanation of this difference? Did they compare the treatment protocols of the Canadian study and their own? Are there variations in the protocols that may explain this difference?
